# Revolutionizing Treatment: Breakthrough Approaches for BCG-Unresponsive Non-Muscle-Invasive Bladder Cancer

**DOI:** 10.3390/cancers16071366

**Published:** 2024-03-30

**Authors:** Maciej Jaromin, Tomasz Konecki, Piotr Kutwin

**Affiliations:** 1st Department of Urology, Medical University of Lodz, 93-513 Lodz, Poland; maciej.jaromin@stud.umed.lodz.pl (M.J.); tomasz.konecki@umed.lodz.pl (T.K.)

**Keywords:** NMIBC, conservative treatment, BCG instillations, BCG-unresponsive, Adstiladrin, Pembrolizumab, TAR-200, N-803, immunotherapy, chemotherapy

## Abstract

**Simple Summary:**

Bladder cancer is a common disease in urological patients. The approach to treatment depends on the severity of the tumor; in this article, we focus on tumors that do not invade the muscle layer of the bladder. Those tumors are resected during an endoscopic procedure (TURBT), but often reoccur. Treating bladder cancer with drugs instead of surgical removal of the bladder (radical cystectomy) is paramount for patients quality of life and overall well-being. The aim of this paper is to review methods of conservative treatment of tumors unresponsive to the typical treatment of choice (BCG instillations).

**Abstract:**

Bladder cancer is the 10th most popular cancer in the world, and non-muscle-invasive bladder cancer (NMIBC) is diagnosed in ~80% of all cases. Treatments for NMIBC include transurethral resection of the bladder tumor (TURBT) and intravesical instillations of Bacillus Calmette-Guérin (BCG). Treatment of BCG-unresponsive tumors is scarce and usually leads to Radical Cystectomy. In this paper, we review recent advancements in conservative treatment of BCG-unresponsive tumors. The main focus of the paper is FDA-approved medications: Pembrolizumab and Nadofaragene Firadenovec (Adstiladrin). Other, less researched therapeutic possibilities are also included, namely: N-803 immunotherapy, TAR-200 and TAR-210 intravesical delivery systems and combined Cabazitaxel, Gemcitabine and Cisplatin chemotherapy. Conservative treatment and delaying radical cystectomy would greatly benefit patients’ quality of life; it is undoubtedly the future of BCG-unresponsive NMIBC.

## 1. Introduction

According to data from the International Agency for Research on Cancer, in 2020, bladder cancer was ranked 10th most popular cancer in the world—6th in the male population and 15th in the female population. In urology, the bladder is the second most popular location for cancer (after the prostate), accounting for 19% of all diagnosed cancers [1]. It is much more prevalent in men, with only ~20% of overall cases diagnosed in the female population [2]. The most common type of neoplasm in the bladder is Urothelial Carcinoma, which accounts for 90% of all tested specimens [3]. Histopathologically, the most common type is non-muscle-invasive bladder cancer (NMIBC), which contributes to 80–75% of all newly diagnosed cases when detrusor muscle is present in a specimen acquired through TURBT [4].

NMIBC types (according to TNM classification) include Ta, T1 and carcinoma in situ (CIS), regardless of WHO histopathological classification (LG/HG). T2, T3 or T4 tumors are muscle-invasive and are treated differently. According to EAU guidelines, the preferred treatment of NMIBC is a complete resection of the tumor by Transurethral Resection of Bladder Tumor (TURBT) with a mandatory histopathological evaluation of the tumor specimen. A second TURBT, or re-TURBT, is recommended after an incomplete first TURBT if there is no detrusor muscle in the specimen and in all T1 tumors. Non-muscle-invasive bladder cancers have a high recurrence rate, estimated at 30–50% after initial TURBT [5].

After histopathological evaluation and full tumor resection, patients should remain under surveillance and should have proper follow-up treatment, depending on their histopathological diagnosis. EAU guidelines include various disease management methods, such as Office Fulguration and Active Surveillance (both in Ta LG/G1 tumors), Adjuvant Intravesical Chemotherapy or Radical Cystectomy. Nevertheless, the golden standard for post-TURBT treatment of NMIBC are Bacillus Calmette-Guérin (BCG) intravesical instillations.

## 2. Intravesical BCG Therapy

BCG therapy in urology has an impressive history—the protocol for BCG intravesical installations created in 1972 by Alvaro Morales is used to this date [6,7]. It is proven that combined therapy is superior to TURBT alone as well as TURBT with intravesical chemotherapy, making TURBT with BCG follow-up the golden standard in NMIBC treatment [8,9,10]. BCG instillations are aimed at stopping recurrences or progression of NMIBC after TURBT, mainly in patients with intermediate- and high-risk tumors as well as CIS tumors. For patients with low-risk tumors, BCG therapy is not recommended due to the very low risk of disease progression; office fulguration is a good alternative [11].

Optimal BCG therapy consists of two periods: induction and maintenance. Induction is divided into six weekly BCG intravesical instillations and then three weekly instillations at 3, 6 and 12 months (each installation usually consists of 81 mg of BCG). Other scheduling variants were tested and proved less effective [12,13]. A one-year installation regiment is recommended for patients with intermediate-risk tumors, while for high-risk tumors maintenance may be prolonged to 3 years. Patients included in the very high-risk group are usually offered immediate radical cystectomy, but BCG therapy is recommended if they are unfit or do not consent to the surgical procedure.

EAU defines BCG failure as the detection of any HG tumor during or after BCG therapy, although the definition is not entirely clear and is used differently in various studies. The failure rate of BCG therapy is estimated to be around 40–50% [14,15]. In cases of unresponsive tumors, the method of choice for further treatment is radical cystectomy. In cases of late relapses (>6 months for T1/Ta HG and >12 months for CIS) and LG/G1-G2 tumors, individual situations should be considered, and usually the BCG treatment is repeated in a 1 to 3 year schedule as presented above. Nonetheless, any recurrences during follow-up visits qualify for radical cystectomy.

There are a few predictors of possible failure of BCG therapy, but most of them are known in patients with previous NMIBC history or after histopathological grading. The neutrophil to lymphocyte (NtL) ratio (measured preoperatively from peripheral blood) is a simple metric that has been proven to be associated with recurrence and progression risks. Elevated NtL prognoses a worse recurrence-free rate and progression-free rate after 5 years of initial TURBT [16,17]. It is not entirely clear what the NtL threshold for unfavorable prognosis should be: both NtL > 2.2 and >2.5 were proved to be associated with worse BCG therapy outcomes [18,19].

The tumor microenvironment is a major prognostic factor for BCG response. BCG instillations generate a local immune response by binding to the fibronectin attachment protein (FAP) of the urothelium, activating macrophages and neutrophils through Toll-like receptors (TLR). Activated macrophages promote the secretion of nitric oxide and further recruitment of immune response cells by cytokine signaling pathways. Neutrophils secrete TNF-related apoptosis-inducing ligand (TRAIL), which shows anti-tumor properties; responders to BCG therapy have higher TRAIL urine levels than non-responders [20,21]. High levels of cytotoxic CD8^+^ T cells, especially their intratumoral presence, predict a good anti-tumor response for BCG and immune checkpoint inhibition. Myeloid-derived suppressor cells in the tumor microenvironment inhibit the proliferation of CD8^+^ T cells; therefore, their low levels are beneficial to BCG therapy outcomes [22]. The stimulator of interferon genes (STING) protein creates an immunostimulatory environment by recruiting M1 macrophages, effector T cells and T-helper type-1 cells. cGAS-STING is another immune response pathway activated in the tumor microenvironment by BCG. Tumors responding to BCG therapy have higher STING protein expression at baseline, further increased by BCG instillations. In higher-grade tumors, expression of STING protein was decreasing (TaLG—22%, TaHG—15%, T1HG—6.5%), implicating tumor resistance to immune response [23].

BCG therapy may cause adverse effects, either local or systemic. Local side effects include cystitis, epididymo-orchitis, symptomatic granulomatous prostatitis and hematuria. Systemic side effects include arthritis, fever > 38.5 °C for >48 h, BCG sepsis or allergic reactions. Interestingly, an EORTC genito-urinary cancer group study of 1355 patients with a median follow-up time of 7.1 years did not show increased toxicity of ⅓ dose compared to full dose but showed that reduced dose produces suboptimal results [24]. Regarding side effects, the same study reported that 62.8% of patients presented local side effects and 30.6% presented systemic side effects—nonetheless, only 7.8% of patients stopped the BCG therapy due to side effects [25]. Another trial, in which 487 patients received BCG therapy, reported that 20% of patients abandoned the treatment due to side effects [26]. Quinolones are proven to be effective at mitigating side effects after BCG instillations, but they may cause damaging and potentially permanent side effects. The use of quinolones in BSC side effects is still being debated and is not a standard procedure [27].

## 3. Radical Cystectomy

NMIBC progression to Muscle Invasive Bladder Cancer varies depending on the tumor’s risk group. In very high groups, percentage rates are very high: 1 year’s progression probability is 16%, 5 years—40% and 10 years—53% (according to WHO 2004/2016 risk groups) [28]. Often, T1 tumors are upstaged to muscle-invasive after Radical Cystectomy—a retrospective, multicenter study of 64,675 patients showed a median upstaging rate of 49.7% [29]. Therefore, after diagnosis of a very high-risk tumor, EAU guidelines recommend immediate Radical Cystectomy, and suggest discussing immediate Radical Cystectomy in high-risk group patients. Oncologically, Radical Cystectomy is a very effective procedure, but it comes with the risk of many adverse effects and, in some cases, might be an overtreatment.

A meta-analysis of adverse effects after Radical Cystectomy concluded that the most common are gastrointestinal complications (20%), infectious complications (17%), ileus complications (14%), cardiovascular complications (9%) and respiratory complications (7%). Similar results were reported in a systematic review [30]. Results of a cohort study reporting on 90-day complications post-Radical Cystectomy showed 24.3% of patients exhibited major (Clavien–Dindo grade 1 or 2) complications, and 53.9% exhibited minor (Clavien–Dindo grade 3, 4 or 5) complications. The re-admission rate was 29.6%, and ER visits were 37.9%, both mostly due to infections. Ninety-day mortality was assessed at 2.1% for 30-day mortality and 4.7% for 90-day mortality [31,32]. The overall survival (OS) and cancer-specific survival (CSS) rates vary depending on the examined population and experience of the institution, but they oscillate between 60–30% for OS and 70–40% for CSS [33,34,35].

Radical Cystectomy greatly influences QoL. A prospective study on a group of 842 patients concluded that 1 year after radical cystectomy, 43.1% of patients suffered from high psychological distress (measured by the Questionnaire on Stress in Cancer Patients). Patients also report lowered QoL due to body image, physical function, constipation, diarrhea, financial situation and many more; specific results vary depending on the method of bladder reconstruction. During 12 months of observation after radical cystectomy, patients’ overall QoL improved, which may indicate further progress after this period [36]. In a multinational experiment, an online questionnaire was submitted to 107 patients with NMIBC, showing hypothetical scenarios for disease management. The goal of the study was to analyze attitudes toward radical cystectomy. Results show that patients would accept an increased risk of progression to MIBC and serious side effects (43.8% and 66.1%, respectively) for delaying radical cystectomy for 5 years.

Although such experiments have many limitations, these data clearly shows patients unwillingness to undergo Radical Cystectomy and is aligned with the clinical experience of many physicians. The possible side effects and stress connected with radical cystectomy lead patients to refuse the operation; far more Radical Cystectomies are canceled due to patients lack of consent than medical contraindications. Even if the operation is unavoidable, most patients will choose some form of conservative treatment to delay the procedure, which should improve their QoL and delay complications related to Radical Cystectomy.

## 4. New Treatment Options

Considering NMIBC treatment, there has been a void between BCG instillations and radical cystectomy, effectively leaving patients with BCG-unresponsive tumors with only one therapeutic option. In recent years, many new approaches and substances have been developed and studied in efforts to enrich standard NMIBC treatment with new non-invasive options. All clinical trials included in this article are listed in Table 1.

## 5. Intravesical Therapies

### 5.1. N-803

N-803, also known as ALT-803, is an immunomodulating drug designed for intravesical use. Pharmacokinetically, it is a superagonist of IL-15 receptors—most importantly in CD8^+^ T and NK cells, promoting their proliferation and activation without broad activation of other proinflammatory pathways [37,38]. Results from a recent clinical trial performed on 160 BCG-unresponsive participants (83 with CIS and 77 with papillary tumors) show great synergistic effects of N-803 and BCG, with a 71% complete response rate for CIS patients in 24.1 months median complete response time. Patients with papillary tumors had a 48% disease-free rate after 24 months. More than 90% of the patients avoided cystectomy two years after BCG + N-803 combined therapy, and none of the patients had any serious adverse effects [39]. Further clinical trials of the BCG + N-803 combination are estimated to end in December 2027 and October 2028.

### 5.2. Nadofaragene Firadenovec 

Nadofaragene firadenovec (or Adstiladrin) is a gene therapy based on non-replicating viral DNA vectors encoding interferon-α2b, promoting the production of interferon-α2b in tumor cells and effectively resulting in apoptosis. Other anti-tumor mechanisms include inducing MHC-I expression, which promotes T cell activity against tumor cells. Adstiladrin is administered intravesically; results from clinical trials (NCT01687244) suggest a recommended dosage of 3 × 10^11^ viral particles/mL in 75 mL, instilled every 12 weeks [40,41]. Some studies look at factors that predict Adstiladrin response durability. The type I Interferon signaling pathway, cytotoxic response to Interferon-α and transcriptional response might be reliable predictors of tumor resistance to Adstiladrin. After administration of Adstiladrin, anti-adenovirus antibody levels may predict the success of further treatment: a titer >800 was observed in 47% of responders compared to 18% of non-responders; it was also associated with a higher probability of a durable response [42,43].

Adverse effects rates of Adstiladrin are comparable to Pembrolizumab: in a phase 3 clinical trial (NCT02773849), 157 patients were treated with BCG-unresponsive NMIBC. Adverse effects were observed in 66% of patients; grade 3 in 4% of patients; no grade 4 or 5 or death connected to treatment was reported; 3 patients discontinued due to adverse effects [44]. Similarly, in the previously mentioned clinical trial, there were no reports of adverse effects graded 4.5 or treatment-related death; none of the participants withdrew from the trial. Both trials concluded that Adstiladrin is well-tolerated and safe to use in the treatment of NMIBC [41].

In clinical trial NCT01687244, 40 patients were divided into two groups: group A received 1 × 10^11^ viral particles/mL every 3 months for 1 year, and group B received 3 × 10^11^ viral particles/mL every 3 months for 1 year. Instillations were administered only to patients with no HG recurrence. Combining results from both groups, 17 patients (42.5%) had no HG recurrence at 9 months (after 3 doses); 14 patients (35%) had no recurrence after 12 months. No patient underwent a cystectomy during the 12-month trial, and the overall survival rate was 100%.

The clinical trial NCT02773849 had a total of 157 patients: group A, including 107 patients with CIS (with or without Ta or T1 tumors), and group B, including 50 participants with Ta HG or T1 without CIS. The dose and regimen of instillations were 3 × 10^11^ viral particles/mL every 3 months for 1 year. In group A, a complete response was achieved by 53.4% and 24.2% of participants after 3 months and 12 months, respectively. In group B, the median recurrence-free survival was 12.35 months. After 24 months, overall survival was 94.4% in group A, and 93.2% in group B; cystectomy-free survival was 64.6% in group A and 69.8% in group B [44,45,46].

Intravesical instillations are a safe option and are usually easy to perform in an office environment. Most importantly, the safety of the aforementioned drugs is comparable to BCG instillations, which are much safer and better tolerated than Radical Cystectomy. Combining new pharmaceuticals with BCG instillations seems to be a very tempting option, enhancing a proven and well-established therapy with tumor-targeting drugs. It is also important to include new diagnostic methods for tumor microenvironment, NtL ratio or genetic profile of tumors to implement the best possible therapy and regiments. Gathering data will help in recognizing best practices and best combinations of drugs, as well as optimal schedules. Whether monotherapy of BCG instillations should remain the gold standard after first TURBT is up for discussion, but intravesical therapy for BCG-unresponsive tumors is definitely the future of NMIBC treatment.

## 6. Intravenous Therapies 

### 6.1. Cabazitaxel, Gemcitabine and Cisplatin

Cabazitaxel, Gemcitabine and Cisplatin (CGC) are well-known chemotherapeutics used in the treatment of many different neoplasms. Cabazitaxel is mostly used in prostate cancer, while Gemcitabine and Cisplatin are implemented in advanced and metastatic bladder cancer [47,48]. However, combining those three in the intravesical therapy of BCG-unresponsive NMIBC is a novel idea, and only a few clinical trials have been made. One trial was based on a method of sequential administrations of CGC weekly during the induction phase and later monthly or bimonthly during maintenance. During Phase I, 18 patients ineligible for radical cystectomy were enrolled and treated with the aforementioned protocol; the recurrence-free rate at 12 months was 83% and at 24 months, 64%, but for patients receiving the maximum CGC dose (10 out of 18), the recurrence-free rate was 100% and 83% at 12 and 24 months, respectively. No dose-limiting toxicities or adverse effects were observed. During Phase II, 31 patients were enrolled; the completion of this trial is expected to be in December 2024 [49,50]. Another study retrospectively reviewed 107 patients who were receiving Gemtabicine and Docetaxel (with similar induction + maintenance protocols) for 2 years. Results show a recurrence-free rate of 85% and 82% at 12 and 24 months, respectively, and no patient had tumor progression or died of bladder cancer [51]. Data for conjoined intravesical chemotherapy of NMIBC are still scarce, but more studies and results may shed light on a new approach to BCG-unresponsive tumor treatment.

### 6.2. Pembrolizumab

Programmed cell death 1 (PD-1) protein is a receptor present in T and B cells, activated by its ligand PD-L1. Its main function is to down-regulate the immune response to humans’ own cells, effectively preventing autoimmune reactions. Stimulating the PD-1/PD-L1 pathway is used by cancer cells to avoid immune response, allowing the progression of tumors [52]. BCG is a nonspecific immunotherapeutic, and combining it with an “immune checkpoint” inhibitor may give effective results even in BCG monotherapy-unresponsive tumors, especially if PD-L1 is present in tumor cells [53].

BCG therapy upregulates PD-L1 expression in tumors in vitro and in rat models, possibly due to tumor infiltration of cytotoxic T-cells during the local immune response. This results in impaired T-cell response through the PD-1/PD-L1 pathway, enabling a synergistic effect with PD-1 checkpoint inhibitors. Combining BCG with anti-PD1 antibodies results in increased T-cell activity; both serum INF-y levels and Granzyme B expression in tumor cells were higher after combined therapy compared to monotherapy and the control group. A reduction of myeloid-derived suppressor cells was also observed [54].

Pembrolizumab is a monoclonal IgG4 anti-PD1 antibody administered intravenously. It is important to note that BCG treatment does not impair the effects of Pembrolizumab. A study on 755 patients with Urothelial Carcinoma (20.5% with previous BCG therapy) concluded that previous BCG therapy did not influence overall survival, disease control rate or objective response rate [55].

There are many ongoing trials testing PD-1 inhibitors: Atezolizumab (NCT03799835, NCT04134000, NCT02792192), Durvalumab (NCT03528694, NCT03759496, NCT04106115), Nivolumab (NCT04149574) Sansanlimab (NCT04165317), and more. Soon we may see PD-1 inhibitors as a staple in conservative treatment of NMIBC. Regarding NMIBC, Pembrolizumab is the only drug accepted by the FDA for the treatment of NMIBC and is going to be the main focus of this whole group. To this date, it has been successfully used in melanoma, non-small cell lung cancer and many more neoplasms.

The KEYNOTE-001 clinical trial enrolled 655 patients with advanced or metastatic melanoma; patients received 2 mg/kg every 3 weeks, 10 mg/kg every 3 weeks, or 10 mg/kg every 2 weeks for at least 6 months in case of a complete response. Treatment was discontinued in cases of patients’ decision to withdraw, intolerable toxicity or disease progression. Treatment-related adverse effects were present in 86% of all patients, but only 17% were grade 3–4, and none experienced treatment-related death [56]. A study on Pembrolizumab effects on non-small cell lung cancer reported a 70.9% rate of all adverse effects in a group of 495 patients. The most common included fatigue, rash, decreased appetite and pruritus. In this group, there were only 9.4% of adverse effects above grade 3 [57]. A meta-analysis of the adverse effects of PD-1 inhibitors (pembrolizumab and nivolumab) on 9136 total patients showed only 33 deaths, the leading cause being pneumonitis [58].

The effects of Pembrolizumab on NMIBC are reported in KEYNOTE-057, a phase 2 single-arm clinical trial. A total of 101 patients from 54 facilities with BCG-unresponsive CIS (63.4% had CIS only, 24.9% had CIS + Ta/HG, 11.9% had CIS + T1) were recruited to Cohort A; patients with no CIS but TaT1 tumors were included in Cohort B. All patients were ineligible or refused RC; it is worth mentioning that the number of patients refusing RC was 95%. Participants were administered 200 mg of intravenous Pembrolizumab every 3 weeks for 2 years. Discontinuation of treatment was caused by tumor resistance, recurrence or progression, toxic effects or patients’ decision to withdraw. The median follow-up was 36.4 months; the median number of Pembrolizumab administrations was 7; and the median time of therapy was 4.2 months.

At 3 months, a complete response was observed in 40.6% of patients; for the CIS-only group, 45%, 0%; for the CIS + Ta/HG group, 29.2%; and for the CIS + T1 group, 41.7%. Progression was reported in 9.4% of patients, and stable disease was reported in 47.9%.

Adverse effects were observed in 61% of the patients; 21% of patients had immune-related adverse effects; and 13% of patients experienced adverse effects grade 3 or above, which is consistent with the adverse effects rate reported in other studies [56,57,58]. There were no treatment-related deaths; 6.9% of patients discontinued the treatment due to adverse effects [59,60]. KEYNOTE-676 is an ongoing, phase 3 clinical trial studying the effects of combining Pembrolizumab and BCG in BCG-naive tumors and BCG-unresponsive high-grade tumors. The study is expected to finish in 2028 [61].

Intravesical instillations of Prembolizumab were tested in two clinical trials. A phase-1 clinical trial (NCT03167151) aimed at studying the safety of intravesical administrations of Prembolizumab, dividing 6 patients into three groups (*n* = 2). Each group received weekly instillations for a total of 6 weeks; each dose was administered in 40 mL infusions retained for 2 h in the bladder. Group 1 started with 50 mg of Prembolizumab in weeks 1 and 2, then proceeded to 100 mg in weeks 3 and 4, and then 200 mg in weeks 5 and 6; group 2 started with 100 mg in weeks 1 and 2 and then 200 mg for weeks 3, 4, 5 and 6; group 3 had a constant dose of 200 mg throughout 6 weeks. During the course of the study, 21 adverse effects were reported, 14 of which were treatment-related. The study does not report any grade 3 or higher adverse effects connected to treatment. Unfortunately, due to delays in recruitment phase 2, the study was abandoned, giving no information on the active intravesical dose of pembrolizumab [62].

The second study inspected the effects of intravesical instillation of Pembrolizumab combined with BCG in nine BCG-unresponsive tumor patients (NCT02808143). The schedule included a pre-induction pembrolizumab instillation (1–2 mg/kg) 2 weeks before the induction phase. During the 5-week induction phase, patients received weekly BCG instillations, combined with Pembrolizumab instillations in weeks 0.2 and 4. After induction, patients received Pembrolizumab instillations every 2 weeks for 8 weeks (4 doses in total), and later Pembrolizumab instillations every 4 weeks for 32 weeks (8 doses in total). There were 22 reported adverse effects, 21 grade 1 or 2 (with 1 incident of serious grade 2 adverse effect) and one grade 5: myasthenia gravis, which resulted in the patient’s death. Overall, only five patients were alive at the end of the study. Among them, the median time of progression-free survival was 35 months. All patients had tumor recurrence, with only one patient having recurrence-free survival for more than a year; progression to muscle-invasive bladder cancer occurred in six out of nine patients [63].

Similarly to intravesical pharmaceuticals, intravenous therapy is a promising tool in NMIBC treatment. Intravenous chemotherapy is a hallmark choice in oncological therapies for many tumors and cancers, including muscle-invasive bladder cancer. In NMIBC, it is not a common choice due to intravesical chemotherapy being less effective in conservative treatment than BCG and less effective than Radical Cystectomy in BCG-unresponsive treatment. Intravenous chemotherapy may seem like an aggressive option for NMIBC, but studies show very good results in recurrence-free and progression-free rates. On the other hand, supporting BCG instillations with “checkpoint inhibitor” immunotherapy seems like a very logical follow-up in patients unresponsive to BCG monotherapy. Intravenous chemotherapy and immunotherapy are likely to become the link between BCG instillations and Radical Cystectomy; similarly to intravesical therapies, more studies and trials are required to establish best practices and regiments.

## 7. Intravesical Delivery Systems

TAR 200 is an intravesical delivery system, releasing gemcitabine in a span of over 2 weeks. In a clinical trial, SunRISe-1 (NCT04640623), TAR 200 alone, TAR 200 + systemic Cetrelimab and systemic Cetrelimab alone were evaluated in BCG-unresponsive NMIBC. The total number of participants is estimated to be 200; the total number of TAR-200 doses is 16. The main objective of the study is the evaluation of the complete response rate and disease-free survival up to 5 years. The estimated completion date is July 2027.

A similar trial (NCT05316155) is using an Intravesical Delivery System to study the proper dosage and adverse effects of Erdafitinib, as well as plasma and urine concentration, recurrence-free survival and complete response. A total of 92 patients are divided into five cohorts, each measured for different outcomes. The study is estimated to finish in June 2027.

Erdafinib is a Fibroblast Growth Factor receptor (FGFR) inhibitor, approved on 12 April 2019 by the FDA for patients with locally advanced or metastatic urothelial carcinoma. The FGFR pathway plays an important role in the apoptosis of some cancer cells; overexpression of FGFR is reported in breast cancer, urothelial carcinomas, melanoma, gastric malignancies and more. FGFR inhibitors are directly impacting the FGF/FGFR pathway but also influence the tumor microenvironment indirectly by altering angiogenesis via VEGF inhibition and local immune responses [64]. In muscle-invasive bladder cancer, FGFR1 mRNA was shown to be a predictor of neoadjuvant cisplatin therapy results. Low FGFR1 mRNA levels were correlated with a higher pathological complete response in tumor cells, measured pre- and post-radical cystectomy. The clinical trial NCT02792192 reported 3% complete response, 37% partial response and 39% stable disease (median progression-free survival of 5.5 months). Notably, FGFR2 and FGFR3 gene mutations are especially relevant; mutations *FGFR2^N549H^*, *FGFR3^Y373C;V555M^*, increase tumor immunity to FGFR-inhibitors [65,66].

Intravesical delivery systems are a novel idea aimed at the slow and constant release of a therapeutic. In theory, a steady influx of drugs may prove crucial in restraining tumor growth and obtaining the maximum effects of administered substances. In addition, patients are exposed to much smaller doses compared to typical intravesical or intravenous therapy, which consequently may limit adverse effects and patient discomfort related to single-dose admission. The results of the above trials are of paramount importance to the future of NMIBC treatment and we will be monitoring them very closely.

## 8. Radiation Therapy

Radiation therapy is commonly used to treat muscle-invasive bladder cancer but is not recommended in NMIBC therapy. A meta-analysis of two clinical trials established that a dose of 55 Gy in 20 fractions should be the standard radiation therapy for locally invasive bladder cancer. Unfortunately, 751 of the 782 recruited patients had muscle-invasive bladder cancer; no information on the isolated NMIBC group is provided [67]. The Chinese trial ChiCTR2200059970 aims at examining the effects of short-course radiotherapy (5 × 5 Gy over 5 or 10 days) combined with a 12-dose intravenous cycle of the PD-1 antibody Toripalimab. The planned patient population is 55 with high/extremely high-risk NMIBC, with or without previous BCG instillation courses. Trial NCT00981656 featured 34 patients with stage I NMIBC; treatment included concurrent three-dimensional conformal radiation therapy (3DCRT) and radiosensitizing chemotherapy (CT) of cisplatin or a combination of mitomycin with 5-fluorouracil. Primary outcome, the 3-year radical cystectomy free rate, was reported to be at 88.2%, but the percentage of alive patients after 3 years was 69.2%, and no data were published on progression to stage 2 or higher. Data on radiation therapy in NMIBC are very scarce and cluttered; more standardized trials and studies are needed.

## 9. Discussion

BCG therapy failure is a turning point in the therapy of NMIBC, leaving patients with few options for conservative treatment. For many years, different methods and ideas were developed, but none proved to be reliable—especially in BCG-unresponsive tumors. Intravesical chemotherapy instillations (either single or recurrent) after TURBT were discussed for years but proved less effective than BCG instillations and not beneficial when combined with BCG instillations [68,69]. Ideas for implementing chemohyperthermia have also been studied, but are not yet proven to be effective and adverse effects may be a limiting factor [70].

Many studies report the tumor microenvironment as an important factor in NMIBC therapy. Local immune response, up- and downregulation of PD1/PDL1, VEGF, FGF/FGFR, cGAS-STING signaling pathways and specific gene mutations all contribute to tumor response to particular treatments. Profiling each tumor by its genetic and immunological properties may enable personalized therapy composed of the most efficient pharmaceuticals and delivery methods for each patient. Tumor microenvironment profiling may prove to be as important as histopathological grading for bladder cancer therapy outcomes; more studies and clinical trials are needed.

Therapies presented in this article may seem groundbreaking and revolutionary, but most of them are either still in the phase of clinical trials or very young and not entirely researched. Other limitations include non-standardized outcome measurements, various inclusion/exclusion criteria (e.g., BCG-failure or BCG-unresponsive tumor definitions), a generally small number of patients in trial groups, and heterogeneity in trials of different pharmaceuticals. Safety of patients is most definitely paramount; nonetheless, treatments described in this article usually passed phase 1 of clinical trials well, and none reported alarming toxicity or abnormal amounts of adverse effects. Allowing patients to decide about risks and trade-offs in their therapy is a hallmark of modern medicine; in NMIBC, the decision between conservative and radical treatment seems to be very personal and hard to navigate. Concerns about the effectiveness and safety of novel therapies should be resolved as soon as possible with high-quality studies and trials; in our opinion, both FDA-approved therapeutics (Adstiladrin and Pembrolizumab) are worth mentioning to patients with BCG-unresponsive NMIBC (Table 2).

## 10. Conclusions

NMIBC treatment is very important in urological oncology since progression to a muscle-invasive tumor results in potential life-threatening complications and quick disease development. Searching for balance between well-tolerated, approachable treatment and satisfying oncological results is the focal point, where new pharmaceuticals and methods should enrich the group of possible choices. For many patients, quality of life and tolerability of the treatment may be as important as clear-cut medical results, leaving the discussion in the realm of percentages, risks and compromises.

A secondary, but ever-important, issue is cost effectiveness. A cohort simulation showed positive results of cost effectiveness with a quality-adjusted life-year ratio [71], but it is far from obvious in different healthcare systems and populations. The price of a single Pebmrolizumab’s dose (lasting 3 weeks) is over $11,000, while one year of Adstiladrin treatment costs $240,000. For most people, such expenses are far beyond their reach, so it is up to insurance companies and healthcare systems to develop a reasonable solution.

## Figures and Tables

**Table 1 cancers-16-01366-t001:** Clinical trials for conservative treatment of NMIBC. * trial completed with posted results, ** status unknown.

Trial	Treatment	Estimated Completion Date	Phase	Route of Administration	Number of Patients	Primary Outcome Measures
NCT01687244	Adstiladrin	2016 r. *	Phase 2	Intravesical	40	Dose and safety assessment: 75 mL of rAd-IFN Dose 3 × 10^11^ Vps/mL
NCT02773849	Adstiladrin	2023 r. *	Phase 3	Intravesical	157	Complete response after 3 months: 53.4%
NCT03167151	Pembrolizumab	2019 r. *	Phase 1 + 2	Intravenous	6	Safety assessment: 200 mg iv. Dose
NCT02808143	Pembrolizumab + BCG	2023 r. *	Phase 1	Intravenous	9	Safety assessment: 1/2 mg/kg Pembrolizumab + 50 mg BGC
NCT04640623	Cetrelimab	2027 r.	Phase 2	Intravesical	200	Complete response and disease-free survival: results not posted
NCT05316155	Erdafitinib	2024 r.	Phase 1	Intravesical	112	Safety assessment: Results not posted
NCT03799835	Atezolizumab	2028 r.	Phase 3	Intravenous	516	Recurrence-free survival after 24 months: results not posted
NCT04134000	Atezolizumab	2024 r.	Phase 1	Intravenous	40	Safety assessment of BCG+ atezolizumab: results not posted
NCT02792192	Atezolizumab /Atezolizumab + BCG	2020 r. *	Phase 1/2	Intravenous	24	Complete response after 6 months: 33.3%/41.7%
NCT03528694	Durvalumab + BCG	2025 r.	Phase 3	Intravenous	1018	Disease-free survival: results not posted
NCT03759496	Durvalumab	2022 r. **	Phase 2	Intravesical	39	Safety assessment: results not posted
NCT04106115	Durvalumab	2029 r.	Phase 1 + 2	Intravenous	64	Safety assessment + disease-free survival: results not posted
NCT04149574	Nivolumab	2024 r.	Phase 3	Intravenous	13	Event-free survival: results not posted
NCT04165317	Sasanlimab /sansalimab + BCG	2026 r.	Phase 3	Intravenous	1070	Event-free survival: results not posted

**Table 2 cancers-16-01366-t002:** Comparison of Pembrolizumab and Adstiladrin.

	Pembrolizumab	Adstiladrin
FDA approval date	16 October 2020	16 December 2022
Mechanism	Monoclonal IgG4 anti-PD1 antibody	Viral DNA vectors, encoding Interferon-α2b
Administration method	Intravenous	Intravesical
Schedule	Every 3 weeks	Every 12 weeks
Adverse effects	Grade 1 and 2: ~60%	Grade 1 and 2: 60–70%
	Grade 3: ~15%	Grade 3: 4%
	Severe: ~10%	Severe: none
Complete response	After 3 months: 41%	After 3 months: 53%
	After 12 months: 19%	After 12 months: 24%
Cost per year	$187,000	$240,000

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
