# Peer review of "Revolutionizing Treatment: Breakthrough Approaches for BCG-Unresponsive Non-Muscle-Invasive Bladder Cancer"

_cancers, 2024, doi:10.3390/cancers16071366_

Round 1
Reviewer 1 Report
Comments and Suggestions for Authors
Authors present an interesting however incomplete overview of current trials on BCG unresponsive Non-Muscle-Invasive Bladder Cancer. However, there are some limitations. I would advise a major revision.
The limitations should be stated clearly. The included studies are heteregoneus and inconsistent as to definition of BCG failure, including criteria, outcome definition.
Please add in the table 1 the information on included patients, treatment and if available outcome.
Table 2 makes no sense as complete different populations and treatments are being compared.
Probably it would make sense to focus the review on instillation therapies and devices first.
The underlying immunonologic mechanism of TME (tumor micro environement) should be taken into consideration and included in the review. BCG infiltrates the urothelium and is taken up by macrophages, initiating local immune activation. A cascade of pro-inflammatory cytokines is activated (Cambier, C.; Takaki, K.K.; Larson, R.P.; Hernandez, R.E.; Tobin, D.M.; Urdahl, K.B.; Cosma, C.L.; Ramakrishnan, L. Myco-422 bacteria manipulate macrophage recruitment through coordinated use of membrane lipids. Nature 2014, 505, 218-222.).
Also, other cells of adaptive immune system like NK cell and different T cells play an important role in immune response to BCG (Ratliff, T.L.; Ritchey, J.K.; Yuan, J.J.; Andriole, G.L.; Catalona, W.J. T-cell subsets required for intravesical BCG immunother-471 apy for bladder cancer. J Urol 1993, 150, 1018-1023, doi:10.1016/s0022-5347(17)35678-1.) . What are the immunologic BCG resistance mechanisms? What is the role of PDL-1 status and T-cell?
Examination of tissues from pre- and post BCG bladder samples revealed that about 30% of patients who did not respond to BCG treatment exhibited intratumoral overrepresentation of PD-L1 at baseline, and T cell shift (Wang, Y.; Liu, J.; Yang, X.; Liu, Y.; Liu, Y.; Li, Y.; Sun, L.; Yang, X.; Niu, H. Bacillus Calmette-Guérin and anti-PD-L1 combi-516 nation therapy boosts immune response against bladder cancer. Onco Targets Ther 2018, 11, 2891-2899, 517 doi:10.2147/ott.S165840.). BCG responders had very low PDL-1 levels. Moreover BCG instillation appears to stimulate the expression of PD-L1 in tumor and inflammatory cells through the induction of T cells.
How many of the included PDL-1 as biomarker for BCG response?
NCT03759496 , NCT05120622 and NCT04106115 (Durvalumab combinations) and NCT02792192 (Atezolizumab) studies are missing.
TAR-210 is delivering Erdafitinib, please add some background information and the value of predictive FGFR-testing: Mayr R, Eckstein M, Wirtz RM, Santiago-Walker A, Baig M, Sundaram R, Carcione JC, Stoehr R, Hartmann A, Bolenz C, Burger M, Otto W, Erben P, Breyer J. Prognostic and Predictive Value of Fibroblast Growth Factor Receptor Alterations in High-grade Non-muscle-invasive Bladder Cancer Treated with and Without Bacillus Calmette-Guérin Immunotherapy. Eur Urol. 2022 Jun;81(6):606-614
Ecke TH, Voß PC, Schlomm T, Rabien A, Friedersdorff F, Barski D, Otto T, Waldner M, Veltrup E, Linden F, Hake R, Eidt S, Roggisch J, Heidenreich A, Rieger C, Kastner L, Hallmann S, Koch S, Wirtz RM. Prediction of Response to Cisplatin-Based Neoadjuvant Chemotherapy of Muscle-Invasive Bladder Cancer Patients by Molecular Subtyping including KRT and FGFR Target Gene Assessment. Int J Mol Sci. 2022 Jul 18;23(14):7898.
Author Response
Dear reviewers,
We applied amendments accordingly to your suggestions. All of the changes in the text are highlighted in green. Regardning your review reports:
Review report 1:
- Limitations are accentuated in the discussion section, as well as in some paragraphs
- Table 1 is updated with new trials, number of participants and primary outcome measurement
- Table 2 is meant to represent the present choice for patients and physitians in bladder-sparing, FDA-approved therapies. It is purposefully included in the discussion to be interpreted subjectively by the reader and show that the choice of conservative therapy is still scarce.
- Intravenous and intravesical therapies are still more developped and researched than intravesical devices and those paragraphs have similar length.
- TME is described in multiple paragraphs and added to discusion
- PDL-1, its role in TME and its relevance for treatment are added multiple paragraphs
- Mentioned trials are added to text and table 1
- FGF/FGFR pathway is discussed in detail in added paragraph
Reviewer 2 Report
Comments and Suggestions for Authors
Dear Author
The manuscript is interesting. The selection of headings and sub-headings are perfect. selected It needs a conclusion to represent an overall finding of the manuscript.
Sincerely
Comments on the Quality of English Language
Minor revision
Author Response
Dear reviewers,
We applied amendments accordingly to your suggestions. All of the changes in the text are highlighted in green. Regardning your review reports:
Review report 2:
1. We updated the discussion to better represent the general tone of the paper
Reviewer 3 Report
Comments and Suggestions for Authors
Manuscript entitled "Revolutionizing Treatment: Breakthrough Approaches for BCG-Unresponsive Non-Muscle-Invasive Bladder Cancer"
This work is superficial and not sound. I would suggest the authors revise this work critically.
1. The authors should discuss the mechanism of each therapeutic strategy in more detail.
2. For drugs with similar mechanism (such as immunotherapy), discussion should be made for the difference.
3. In the table, the authors should provide the response rate and the selection biomarkers for each trial and also the enrolled case number.
4. The role of radiation therapy for NMIBC should also be mentioned.
Comments on the Quality of English Language
Revision required
Author Response
Dear reviewers,
We applied amendments accordingly to your suggestions. All of the changes in the text are highlighted in green. Regardning your review reports:
Review report 3:
1. We discussed molecuar mechanisms in more detail in multiple paragraphs, including more information of TME
2. We did not find any metaanalysis compating different bladder sparing treatments in NMIBC; furthermore, there is a vast difference in outcome measures and heterogeneity of trials. Could you specify which differences you would like to be compared?
3. Table 1 is updated; unfortunetely, response rates are not included in all trials
4. A whole section for radiotherapy is added.
Round 2
Reviewer 3 Report
Comments and Suggestions for Authors
The article is well-revised and is accepted in its present form.
Comments on the Quality of English Language
Acceptable